# Changes in Volatile Composition of Cabernet Sauvignon (*Vitis vinifera* L.) Grapes under Leaf Removal Treatment

**Zhiyu Li, Dongyue Yang, Xueqiang Guan, Yuxia Sun and Junfang Wang ***

Shandong Academy of Grape, Shandong Engineering Technology Research Centre of Viticulture and Grape Intensive Processing, Winegrape and Wine Technological Innovation Center of Shandong Province, Jinan 250100, China; lizhiyu1008@163.com (Z.L.); yangdongyue1203@163.com (D.Y.); guanxq90@126.com (X.G.); sunyuxia1230@163.com (Y.S.)
* Correspondence: jfwang1985@126.com

**Abstract:** Several studies have revealed that fruit-zone leaf removal could change the microclimate of grapevine growth, thereby causing complex effects on fruit composition. This study analyzed the profiles of volatiles in Cabernet Sauvignon (*Vitis vinifera* L.) grapes exposed to leaf removal treatment at different phenological periods in three continuous years. The treatments (leaf removal before flowering, after flowering, and veraison) were applied to Cabernet Sauvignon grapevines grown in Yantai (Shandong, China). Berry samples were harvested at maturation to determine the chemical composition, including total soluble solids, phenols, and volatiles. Leaf removal (particularly before flowering) could increase total soluble solids and phenols (anthocyanins, flavonols, and tartaric esters). Volatiles greatly changed in the different years, and leaf removal before flowering could increase the concentration of amino acid-derived volatiles and isoprene-derived volatiles compared with leaf removal after flowering or at veraison. This research provides a basis for further studies on optimizing Cabernet Sauvignon aroma and breeding in vineyards.

**Keywords:** aromatics; terpenes; leaf removal; *Vitis vinifera* L.

## 1. Introduction

Cultivation measures have been studied to improve grape quality [1]. Factors such as soil [2], climate [3], light [4], and cultivation management [5] could change the microclimate of grapevine growth, thereby impacting its quality and causing complex effects on fruit quality. Shaping, pruning, cover cropping, water deficiency [6], and leaf removal [7,8] are the commonly used practices in vineyards.

As a common cultivation practice, timely and appropriate leaf removal treatment could improve the microclimate of grapevine, thereby affecting the composition of grape fruits and ultimately improving the quality of wine [9,10]. By applying leaf removal, canopy density can be controlled and reduced to increase light exposure and change the microclimate of the fruit zone, thereby improving berry quality. Leaf removal can be applied from flowering to veraison [11–13]. Early leaf removal (before flowering) can regulate yield components and improve grape quality by reducing fruit set and berry weight, thereby resulting in smaller, less compact clusters and higher skin–berry ratio [14]. In addition, the decrease in cluster compactness leads to the decrease in cluster humus incidence. This result can be attributed to improved ventilation in sparse clusters [15–17]. Previous studies have shown that leaf removal can exert positive effects on soluble solids and anthocyanins of grapes [18,19], and leaf removal before flowering has an evident effect on improving the soluble solid content and aroma component of grape fruits and the content of phenolic substances [20].

In addition, the volatile composition of grapes plays an important role in the quality of grape and wine because they can reflect the characteristics of grape varieties, regions, and years [21,22]. In previous studies, leaf removal could produce high glycoside precursors in

Riesling [23], and terpenes were significantly increased in Sauvignon Blanc, resulting in the floral fragrance and tropical fruit flavor in wine [24]. However, most previous studies focused on leaf removal at a single period, and few results were reported on leaf removal at different phenological periods.

Yantai is an important wine region in China, where wine grapes and wines have been produced since the 19th century. However, compared with other wine regions in the world, Yantai endures a continental climate characterized with less sunshine, high temperature, and high humidity during the berry development period, resulting in less flavor substances in grape and wine [25]. Many kinds of viticulture strategies (including leaf removal) are suggested and tested in vineyards of Yantai to reduce this negative effect. Thus, this study aimed to evaluate the effect of leaf removal at different phenological stages on volatile concentration and profiles in Yantai, thereby exploring the possibility of improving wine aroma by applying leaf removal in this continental-climate wine region.

## 2. Materials and Methods

### 2.1. Field Experiment and Sample Preparation

The field trial was conducted over the growing seasons in 3 years (2017, 2018, and 2019) at a commercial vineyard (120°89′68″ east latitude, 37°77′29″ north latitude, COFCO, Great Wall Winery, Shandong, China). Nine-year-old Cabernet Sauvignon (*Vitis vinifera* L.) vines were grafted on SO4 rootstock and planted in north–south-oriented rows at a spacing of 2.5 m × 1 m. The vines were single cordon-trained and 8–12 two-bud spurs per vine were retained. Clean-tillage was implemented in the vineyard soils, and organic fertilizer was applied every winter and spring.

Leaf removal was established and initialed 10 days before flowering (LR-BF), 35 days after flowering (LR-AF), and during veraison (LR-V). Vines without leaf removal served as the control (CN). Leaf removal was performed manually in the cluster zone, wherein all primary and lateral leaves were removed from the basal node to the third node (Supplemental Figure S1).

Weather data (from April to October) were collected from a meteorological weather station located in the vineyard (Table 1). Each treatment was replicated five times in plots of 10 vines based on a randomized block design. Ten clusters were randomly collected from each plot at harvest, frozen on dry ice, and transferred to a laboratory immediately, where they were stored at −40 °C.

**Table 1.** Temperature, precipitation, sunshine duration and relative humidity from April to October in three years.

| Year | Month | Temperature (°C) | | | Precipitation (mm) | Sunshine Duration (h) | Relative Humidity (%) |
|---|---|---|---|---|---|---|---|
| | | Min. | Max. | Avg. | | | |
| 2017 | April | 8.3 | 18.1 | 12.8 | 42.2 | 242.9 | 54 |
| | May | 14.4 | 23.8 | 18.8 | 27.7 | 275.0 | 52 |
| | June | 18.8 | 27.5 | 22.8 | 27.3 | 269.1 | 64 |
| | July | 22.1 | 29.3 | 25.3 | 34.9 | 250.6 | 69 |
| | August | 22.3 | 29.3 | 25.7 | 276.1 | 223.2 | 74 |
| | September | 18.0 | 25.2 | 21.5 | 106.7 | 216.1 | 68 |
| | October | 13.1 | 20.1 | 16.3 | 18.2 | 183.9 | 60 |
| 2018 | April | 9.7 | 19.7 | 14.3 | 37.0 | 251.1 | 50 |
| | May | 13.8 | 24.1 | 18.7 | 58.1 | 237.1 | 55 |
| | June | 19.2 | 28.2 | 23.4 | 71.7 | 265.3 | 62 |
| | July | 22.8 | 30 | 25.9 | 130.4 | 227.5 | 77 |
| | August | 20.7 | 27.1 | 23.9 | 129.0 | 210.5 | 63 |
| | September | 18.2 | 25.4 | 21.6 | 25.6 | 225.6 | 72 |
| | October | 12.5 | 19.3 | 15.5 | 35.1 | 162.8 | 69 |
| 2019 | April | 9.9 | 19.4 | 14.3 | 20.8 | 240.5 | 52 |
| | May | 15.1 | 26.4 | 20.4 | 19.9 | 323.7 | 52 |
| | June | 18.8 | 28.1 | 23.1 | 74.5 | 279.5 | 58 |
| | July | 23.8 | 30.6 | 26.9 | 159.1 | 176.6 | 80 |
| | August | 22.7 | 28.8 | 25.5 | 195.3 | 221.2 | 81 |
| | September | 18.7 | 27 | 22.4 | 14.8 | 288.3 | 71 |
| | October | 11.7 | 18.4 | 14.8 | 32.5 | 162.5 | 65 |

## 2.2. General Grape Parameters

Ten randomly picked grape clusters were collected at every year harvest from each treatment, transported to the lab refrigerators and kept $-40\ °C$ until analysis. The weights of clusters and berries were measured with WPX 4500 (RADWAG, Radom, Poland) electronic scales (0.1 g accuracy). Total soluble solid (TSS) content of grapes was determined as Brix, using a PAL-1 (Atago, Tokyo, Japan) refractometer. The pH of the must was measured using a pH-meter (Elmetron 501, Zabrze, Poland). Titratable acidity (TA) was determined via titration of a water extract of juice with 0.1 N NaOH to an end point of pH 8.1 [26].

## 2.3. Phenolic Compound Analysis

Total phenols, anthocyanin, flavonols, and tartaric esters were detected using the method described by Hose et al. [27]. A total of 1 g of lyophilized and ground grape skins was extracted in 10 mL of extraction solution (80% acetone with 1% formic acid) and sonicated for 30 min at 30 °C. Extracts were then centrifuged at $13,000\times g$ for 10 min at 20 °C. The supernatant was collected and dried using a rotary evaporator, then redissolved in 10 mL of 12% ethanol (with 5 g/L of potassium hydrogen tartrate), and adjusted to pH 3.3 using HCl.

The extracted liquid was diluted 2.5 times for detection. The reaction system was as follows: 4.55 mL of 2% HCl + 0.25 mL of 95% ethanol containing 0.1% HCl + 0.25 mL of extract. After standing for 15 min in the absence of light, absorption values were measured using a UV spectrophotometer at 280, 320, 360, and 520 nm. A280, A320, A360, and A520 correspond to total phenol content, tartarate, flavonol, and anthocyanin, respectively. The calibration curve of total phenols, tartrate, flavonols and anthocyanins were obtained using 0~100 µg/mL gallic acid, caffeic acid, quercetin and malvacin-3-glycoside, respectively.

## 2.4. Volatile Compound Analysis

Volatile compounds were extracted from berry samples using headspace-solid phase microextraction (HS-SPME). Skins and pulps were separated from seeds and ground into juice using a sterile mortar, where calcium chloride ($CaCl_2$, Sigma-Aldrich, St. Louis, MO, USA) was added to increase ionic strength. For headspace sampling, 8 mL of grape juice and 1.5 g NaCl was added in a 20 mL of SPME glass screw cap vial with 20 µL of internal standard (4-methyl-2-pentanol; the final concentration was 20 µg/L in the vial).

Analyses were performed using an Agilent 7890B gas chromatograph coupled to an Agilent 5977A mass spectrometer (Agilent Technologies Inc., Santa Clara, CA, USA). Prior to injection, samples were pre-incubated at 40 °C for 10 min and stirred at 250 rpm. After agitation, a DVB/CAR/PDMS fiber was injected into the vial headspace for volatile absorption for 50 min. Volatiles were thermally desorbed for 5 min at 250 °C. Volatiles were separated with a 30 m × 250 µm × 0.25 µm DB-WAX column (Agilent Technologies Inc., Santa Clara, CA, USA). The oven-temperature program was as follows: hold at 50 °C for 1 min, then heat up to 220 °C at a rate of 1 °C, and hold at this temperature for 5 min. The interface temperature of mass spectrometry was 280 °C; the ion source temperature was 230 °C; the ionization mode was the electron impact ion source; the ion energy was 70 eV; and the mass scanning range was 19–350 ($m/z$). The column flow rate was 0.8 mL/min, and different compounds were separated via programmed heating. The carrier gas was helium.

The identification of VOCs was carried out using MSD Chemstation (E.15.0.0.498, Agilent Technologies Inc., USA) according to Wang et al. [5]. Dominant ions were used to select peaks, and only VOCs peaks with signal-to-noise ratios greater than 10:1 were considered for data quantification. Identification was achieved by comparing the mass spectra with those of authentic standards when available, and with MS from the data system library (Nist2014, Agilent Technologies Inc. USA) and previous published retention indices.

The calibration curves were constructed according to Wang et al. [5]. Seven concentrations of the mix standard solutions were injected in triplicate. Standard calibration curves of 25 compounds (1-propanol, 1-heptanol, 1-pentanol, 1-nonanol, 2-octanol, hexanal, 2-hexenal, 2-octanone, acetic acid, 2-methyl-1-propanol, 2-ethyl-1-hexanol, benzyl alcohol,

acetic acid, hexyl ester, acetic acid, 2-phenylethyl ester, acetic acid, methyl ester, hexanoic acid, ethyl ester, geraniol, linalool, citral, D-limonene, and α-terpineol) were constructed. Independent stock solutions (in ethanol) were prepared for each standard. Standard solutions of the compounds were prepared by diluting the original stock solution of mixtures with simulated wine solution in order to obtain a range of concentrations (0~500 µg/mL). 4-methyl-2-pentanol was used as the internal standard (IS). Concentrations were calculated using the calibration curves of the available standard with the closest chemical structure. Semi-quantification was carried out for the compounds for which standards were not available; all reagents and standards were purchased from Sigma-Aldrich (St Louis, MO, USA).

*2.5. Statistical Analysis*

All data were presented as mean values and their standard errors. A one-way ANOVA with the least significant different (LSD) test used to examine was performed using SPSS 20.0 program (IBM, Armonk, New York, NY, USA) to detect significant differences ($p < 0.05$) among leaf removal treatments at each year for all parameters considered. Factor analysis was performed (SPSS) to determine the relationship among different variables and select the principal component analysis (PCA) analysis to indicate the relationship between volatile compounds and leaf removing in different years.

## 3. Results and Discussion

*3.1. Environmental Conditions and Fruit Zone Microclimate*

The average values of air temperature, rainfall summation, sunshine duration, and medial humidity were calculated from 1 April to 15 October (Table 1). The three seasons were similar, but the distribution of these parameters was quite different during the growing period. The average temperature in 2017 was 12.8 °C, which was 1.5 °C lower than the other two years during pre-blooming, and rainfall was reduced every year during this period. During berry set (middle May), the average temperature was similar in three seasons, with 2019 being slightly warmer (20.4 °C) than the other two years. However, May of 2017 was characterized by considerable rainfall, and the air temperature in June and July of 2017 was lower than the other two years. During veraison (late July), the average temperature in 2019 was 1.5 °C higher (26.9 °C) than the other two years, and the rainfall in 2017 was highest amongst the three years.

With regard to the temperature of fruit zone shown in Table 2, hours of temperature above 35 °C were longer during leaf removal in all three years. In addition, the average daily maximum temperature, daily mean temperature, and average daily temperature difference were higher during leaf removal, except for 2018. The average daily minimum temperature in CN and leaf removal was the same.

Light intensity was high during leaf removal in three years (Table 3). Analyzing different qualities, the percentage of ultraviolet, purple, blue, and cyan light of the fruit zone was high during leaf removal. In LR-V, the percentage of green, yellow, orange, and red light was also high.

**Table 2.** Temperature of fruit zone.

| Year | Month | 16 June–15 July | | 16 July–15 August | | 16 August–15 September | | 16 September–15 October | |
|---|---|---|---|---|---|---|---|---|---|
| | | CN | Leaf Removal | CN | Leaf Removal | CN | Leaf Removal | CN | Leaf Removal |
| 2017 | >35 °C (h) | 21 | 38 | 12 | 25 | 0 | 4 | 0 | 1 |
| | Ave. $D_{Max}$ (°C) | 33.84 | 36.65 | 32.57 | 34.31 | 29.92 | 32.48 | 25.81 | 29.71 |
| | Ave. $D_{Mini}$ (°C) | 20.12 | 19.46 | 20.00 | 20.15 | 17.44 | 17.30 | 12.27 | 12.41 |
| | Ave. D (°C) | 26.46 | 27.02 | 24.98 | 25.60 | 22.77 | 23.19 | 18.57 | 19.32 |
| | Ave. $D_{different}$ (°C) | 13.72 | 17.19 | 12.58 | 14.16 | 12.48 | 15.18 | 13.54 | 17.29 |
| 2018 | >35 °C (h) | 25 | 52 | 33 | 35 | 1 | 3 | 0 | 0 |
| | Ave. $D_{Max}$ (°C) | 32.42 | 35.14 | 33.99 | 33.84 | 30.58 | 29.03 | 25.26 | 23.92 |
| | Ave. $D_{Mini}$ (°C) | 20.13 | 19.45 | 23.39 | 22.88 | 18.59 | 18.06 | 12.91 | 12.38 |
| | Ave. D (°C) | 25.52 | 25.75 | 27.55 | 27.36 | 23.55 | 23.13 | 18.26 | 17.81 |
| | Ave. $D_{different}$ (°C) | 12.29 | 15.68 | 10.60 | 10.96 | 11.99 | 10.97 | 12.35 | 11.54 |
| 2019 | >35 °C (h) | 38 | 97 | 0 | 45 | 0 | 32 | 0 | 17 |
| | Ave. $D_{Max}$ (°C) | 32.29 | 37.82 | 29.51 | 35.46 | 27.84 | 35.11 | 24.48 | 29.41 |
| | Ave. $D_{Mini}$ (°C) | 21.05 | 21.42 | 21.65 | 22.69 | 17.41 | 18.70 | 11.70 | 12.23 |
| | Ave. D (°C) | 26.08 | 27.39 | 24.86 | 26.87 | 22.04 | 24.38 | 17.40 | 18.77 |
| | Ave. $D_{different}$ (°C) | 11.24 | 16.40 | 7.86 | 12.77 | 10.43 | 16.41 | 12.79 | 17.18 |

**Table 3.** Light quality of fruit zone.

| | | Wave Range (nm) Light Quality | <390 Ultraviolet | 390~435 Purple | 435~450 Blue | 450~492 Cyan | 492~577 Green | 577~596 Yellow | 597~622 Orange | 622~760 Red | >760 Infrared | Total |
|---|---|---|---|---|---|---|---|---|---|---|---|---|
| LR-BF | CN | Light intensity (Lux) | 32.6 | 158.3 | 94 | 344.9 | 1440 | 4022.4 | 328.7 | 3708.4 | 10,879.9 | 21,009.1 |
| | | Percentage (%) | 0.15 | 0.75 | 0.45 | 1.64 | 6.85 | 19.14 | 1.56 | 17.65 | 51.79 | 100 |
| | Leaf removal | Light intensity (Lux) | 326.8 | 1251.4 | 609.2 | 1602.6 | 2610.5 | 6938.7 | 722 | 6244.7 | 15,100.7 | 35,406.6 |
| | | Percentage (%) | 0.92 | 3.49 | 1.71 | 4.55 | 7.48 | 19.83 | 2.07 | 17.83 | 42.11 | 100 |
| LR-AF | CN | Light intensity (Lux) | 18.63 | 109.29 | 57.85 | 186.79 | 504.69 | 1257.17 | 90.82 | 1169.64 | 3443.02 | 6837.9 |
| | | Percentage (%) | 0.267 | 1.583 | 0.837 | 2.713 | 7.347 | 18.37 | 1.323 | 17.1 | 50.463 | 100 |
| | Leaf removal | Light intensity (Lux) | 449.23 | 1911.38 | 825.64 | 1873.45 | 2660.29 | 7049.86 | 722.13 | 6360.21 | 25,953.71 | 47,805.89 |
| | | Percentage (%) | 0.94 | 3.997 | 1.727 | 3.917 | 5.563 | 14.747 | 1.51 | 13.303 | 54.287 | 100 |
| LR-V | CN | Light intensity (Lux) | 0.55 | 3.99 | 3.31 | 20.57 | 301.3 | 2703.59 | 56.66 | 2654.3 | 6274.91 | 12,019.16 |
| | | Percentage (%) | 0 | 0.03 | 0.03 | 0.17 | 2.51 | 22.49 | 0.47 | 22.08 | 52.21 | 100 |
| | Leaf removal | Light intensity (Lux) | 8.59 | 62.49 | 50.19 | 309.1 | 2682.09 | 12,495.98 | 634.53 | 11,887.13 | 16,865.88 | 44,995.95 |
| | | Percentage (%) | 0.015 | 0.13 | 0.105 | 0.66 | 5.805 | 27.61 | 1.375 | 26.295 | 38 | 100 |

### 3.2. Berry Growth

When wine grapes have a certain content of TSS, they can make excellent wines [28], and appropriate TA content not only balances taste but also constitutes an important part of the wine body [29]. Studies have shown that leaf removal slightly changed the leaf area [20], and still had enough leaves to support the development of berries without affecting the content of TSS and berry ripening [30]. However, based on previous reports, TSS increased after leaf removal [17,18], and TA was greatly influenced by the environment and light [31,32]. Given the berry weight, the treatment showed no effects during the three years (Figure 1). LR-BF could increase the TSS content in grapes during the three years (2017, 2018, and 2019) but decreased the TA level in 2017 and 2019. However, the TA level increased for LR-AF berries in 2018 (6.40 $\pm$ 0.10 g/L) but decreased in 2019 (4.69 $\pm$ 0.10 g/L). Collectively, LR-AF played a role in the decrease in pH in 2017 (3.36 $\pm$ 0.12), significantly lower than CN and the other two treatments. In this study, high TSS and low TA were also observed after leaf removal treatments, particularly in LR-BF, which was consistent with previous research [15,33].

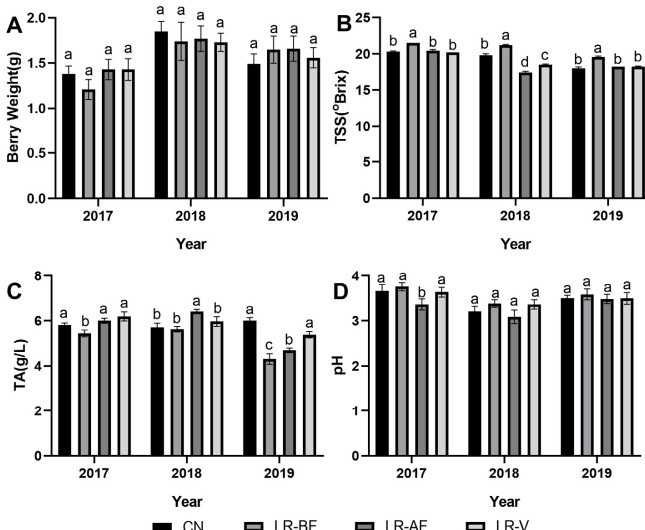

**Figure 1.** Berry weight (**A**), total soluble solids (**B**), titratable acidity (**C**) and pH (**D**) in Cabernet Sauvignon berries exposed to CN, LR-BF, LR-AF and LR-V in three years (2017, 2018, and 2019). LR-BF: leaf removal at 10 days before flower, LR-AF: leaf removal at 35 days after flowers, LR-V: leaf removal at veraison. TSS: total soluble solid content, TA: total titrated acid content. Different letters indicate significant differences ($p < 0.05$) among treatments using one-way ANOVA with the least significant different (LSD) test used to examine the means.

### 3.3. Phenol Contents

Phenols are the important characteristic substances in grape and wine, which play an important role in wine color and astringency [16,34]. Phenol contents were consistent in three years (Figure 2). Anthocyanin, flavanols, tartaric esters, and total phenols were increased for LR-BF and LR-AF berries in 2017 and 2018, except for tartaric esters in 2017. In 2019, only LR-BR showed a high content of anthocyanin, tartaric esters, and total phenols, as well as a high content of total phenols in LR-V. Leaf removal treatments could broaden fruit exposure area and facilitate light absorption on grape berries. Sunlight had a positive effect on the synthesis of phenols in grapes [35–37], and leaf removal treatments improve the photosynthetic capacity of the remaining leaves and induce flavonoid synthesis as a stress response in the common grapevine [38]. In this study, LR-BF and LR-AF increased light exposure and temperature difference on berries, resulting in the accumulation of phenols in grapes. In addition, the total sunshine duration was 873.8, 826.4, and 848.6 h in these three years from July to October (veraison to harvest), respectively, which means that the grapes received the most total sunshine duration during the ripening process in 2017.

That explains the lower amount of anthocyanins, tartaric esters, and total phenols in 2018 and 2019 than in 2017.

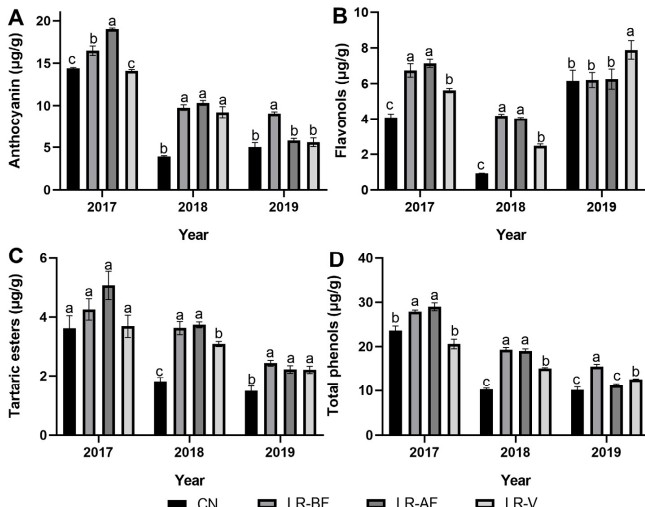

**Figure 2.** The anthocyanin (**A**), flavonols (**B**), tartaric esters (**C**), and total phenols (**D**) concentration in Cabernet Sauvignon berries exposed to CN, LR-BF, LR-AF and LR-V in three years (2017, 2018, and 2019). LR-BF: leaf removal at 10 days before flower, LR-AF: leaf removal at 35 days after flowers, LR-V: leaf removal at veraison. Different letters indicate significant differences ($p < 0.05$) among treatments using one-way ANOVA with the least significant different (LSD) test used to examine.

### 3.4. Profiles of Volatiles

3.4.1. Fatty Acid-Derived Volatiles

Fatty acid-derived volatiles were considered to be the main component in Cabernet Sauvignon berries which caused the unpleasant herbaceous flavor, such as green peppers, in wine [39]. Thirty-two fatty acid-derived volatiles were identified and quantified in Cabernet Sauvignon berries from 2017 to 2019: fourteen straight-chain alcohols, nine straight-chain esters, seven straight-chain carbonyls, and two straight-chain acids (Table 4).

C6 compounds, such as 1-hexanol, 2-hexen-1-ol, hexanoic acid, ethyl ester, and hexanal, were increased after leaf removal treatments in three years (particularly in 2017). Straight-chain alcohols such as 2-Penten-1-ol, 1-Hexanol, and 2-Hexen-1-ol were decreased in 2018 and 2019 after leaf removal in different periods. In addition, straight-chain carbonyls such as 2-Octanone was decreased in 2018 and 3-Octanone was decreased in 2017 and 2018. In the total fatty acid-derived volatiles, there was a significant decrease in 2018 and 2019 compared to CN after leaf removal treatment in different periods. Therefore, leaf removal can reduce the content of C6 compounds to a certain extent, thereby reducing the herbaceous flavor of wine.

1-Hexanol and 2-hexen-1-ol were the major straight-chain aliphatic alcohols identified in 2017 and 2019, while 2-hexen-1-ol in 2018 (66.68 ± 1.83 to 128.75 ± 13.53 µg/L with different treatments) was less than the other two years (1030.78 ± 7.58 to 3591.93 ± 13.22 µg/L in 2017 and 2453.03 ± 11.63 to 4462.67 ± 84.20 µg/L in 2019). Among the straight-chain aliphatic alcohols, 1-hexanol, 2-hexen-1-ol, 1-nonanol, and 2-nonanol were increased in LR-BF and LR-AF berries in 2017. In 2018, 1-pentanol (90.49 ± 3.62 µg/L), 2-penten-1-ol (47.07 ± 0.91 µg/L), 2-nonanol (225.97 ± 9.52 µg/L), 2-octanol (84.81 ± 1.69 µg/L), 1-octen-3-ol (105.08 ± 1.35 µg/L), and 3-nonen-1-ol (58.85 ± 1.96 µg/L) concentration were high in LR-BF and 2-octanol (78.84 ± 1.82 µg/L) was high in LR-AF. Similarly, 2-octanol (72.57 ± 1.32 µg/L), 1-octen-3-ol (122.58 ± 3.35 µg/L), and 3-nonen-1-ol (55.33 ± 3.52 µg/L) were high in LR-V in this year. In addition, a number of straight-chain aliphatic alcohols such as 2-Penten-1-ol, 1-Hexanol, and 2-Hexen-1-ol in 2018 and 2019 were decreased after leaf removal in different period. In 2019, 1-pentanol, 2-nonanol, and (6Z)-nonen-1-ol were increased in LR-BF and LR-V, whereas most alcohols were decreased in 2019.

**Table 4.** Fatty acid-derived volatiles in Cabernet Sauvignon berries exposed to CN, LR-BF, LR-AF and LR-V in 2017, 2018, and 2019.

| Compounds (µg/L) | 2017 | | | | 2018 | | | | 2019 | | | |
|---|---|---|---|---|---|---|---|---|---|---|---|---|
| | CN | LR-BF | LR-AF | LR-V | CN | LR-BF | LR-AF | LR-V | CN | LR-BF | LR-AF | LR-V |
| **Straight-chain alcohols** | | | | | | | | | | | | |
| 1-Propanol | 19.75 ± 1.49 c | 35.87 ± 2.83 a | 22.33 ± 1.47 c | 29.15 ± 1.52 b | 164.95 ± 2.63 a | 160.45 ± 6.41 a | 162.68 ± 6.86 a | 156.93 ± 7.08 a | - | - | - | - |
| 1-Heptanol | 127.88 ± 7.72 a | 125.85 ± 4.15 a | 127.76 ± 1.71 a | 119.07 ± 12.62 a | 102.22 ± 7.88 a | 109.33 ± 6.15 a | 132.80 ± 13.70 a | 118.84 ± 7.91 a | 26.02 ± 0.65 a | 16.64 ± 1.36 b | 20.94 ± 0.94 b | 22.36 ± 0.28 b |
| 2-Heptanol | 72.14 ± 10.17 a | 74.84 ± 2.84 a | 74.6 ± 2.14 a | 77.06 ± 3.20 a | 113.77 ± 8.02 a | 86.41 ± 2.68 b | 110.15 ± 5.44 a | 89.65 ± 2.20 b | - | - | - | - |
| 1-Pentanol | 51.25 ± 7.67 a | 44.39 ± 3.01 a | 33.56 ± 0.86 a | 40.81 ± 8.50 a | 58.18 ± 6.41 b | 90.49 ± 3.62 a | 40.47 ± 0.10 c | 53.14 ± 3.18 b | 14.77 ± 2.15 b | 18.64 ± 0.16 a | 16.24 ± 1.28 b | 17.00 ± 0.64 b |
| 2-Penten-1-ol | 54.36 ± 2.80 a | 56.39 ± 1.99 a | 56.20 ± 2.01 a | 57.04 ± 1.81 a | 68.71 ± 1.52 a | 47.07 ± 0.91 c | 51.48 ± 1.33 b | 46.06 ± 0.70 c | 17.34 ± 0.29 a | 15.67 ± 0.28 b | 15.14 ± 0.68 b | 15.91 ± 0.58 b |
| 1-Hexanol | 6773.1 ± 211.27 c | 9868.53 ± 92.33 a | 8963.71 ± 87.06 a | 8863.62 ± 102.00 b | 8855.95 ± 75.57 b | 9720.69 ± 88.03 a | 7765.65 ± 88.34 c | 7891.73 ± 94.13 c | 7514.45 ± 40.67 a | 4668.4 ± 50.28 d | 5569.75 ± 76.43 b | 5082.95 ± 13.17 c |
| 2-Hexen-1-ol | 2418.74 ± 11.17 c | 2805.73 ± 21.43 b | 3591.93 ± 13.22 a | 1030.78 ± 7.58 d | 128.75 ± 13.53 a | 75.56 ± 1.01 b | 66.68 ± 1.83 b | 71.69 ± 4.33 b | 4462.67 ± 84.20 a | 3217.66 ± 52.23 b | 2865.61 ± 22.48 c | 2453.03 ± 11.63 c |
| 3-Hexen-1-ol | 312.56 ± 12.56 a | 303.11 ± 9.26 a | 305.13 ± 2.53 a | 300.24 ± 2.72 a | 326.18 ± 7.32 a | 334.09 ± 6.33 a | 327.32 ± 2.66 a | 314.56 ± 4.54 a | 118.41 ± 3.36 a | 85.70 ± 5.76 b | 92.68 ± 2.32 b | 89.22 ± 1.65 b |
| 1-Nonanol | 63.18 ± 1.05 b | 133.38 ± 5.98 a | 113.46 ± 1.79 a | 140.22 ± 7.37 a | 441.89 ± 5.92 a | 460.07 ± 5.38 a | 495.21 ± 2.91 a | 263.52 ± 9.04 b | 30.5 ± 1.87 a | 31.77 ± 0.73 a | 28.2 ± 3.10 a | 29.01 ± 1.36 a |
| 2-Nonanol | 27.26 ± 1.83 d | 40.51 ± 2.94 a | 16.45 ± 1.11 c | 26.41 ± 1.46 b | 155.95 ± 2.94 b | 225.97 ± 9.52 a | 175.46 ± 8.11 b | 143.44 ± 8.40 b | 5.53 ± 0.03 c | 9.23 ± 0.85 a | 4.24 ± 0.32 d | 6.37 ± 0.22 b |
| 2-Octanol | 55.41 ± 2.75 a | 49.54 ± 1.39 a | 52.56 ± 2.96 a | 54.26 ± 1.08 a | 41.07 ± 0.40 b | 84.81 ± 1.69 a | 78.84 ± 1.82 a | 72.57 ± 1.32 a | 18.34 ± 1.02 a | 20.53 ± 1.66 a | 16.33 ± 2.10 a | 19.36 ± 1.09 a |
| 1-Octen-3-ol | 136.8 ± 1.72 a | 100.81 ± 8.46 a | 114.56 ± 2.23 a | 98.56 ± 7.75 a | 82.82 ± 2.33 b | 105.08 ± 1.35 a | 95.98 ± 1.55 b | 122.58 ± 3.35 a | 157.58 ± 3.89 a | 105.57 ± 1.97 b | 111.96 ± 3.41 b | 93.25 ± 1.09 c |
| 3-Nonen-1-ol | - | - | - | - | 46.31 ± 3.16 b | 58.85 ± 1.96 a | 50.15 ± 2.22 b | 55.33 ± 3.52 a | 4.24 ± 0.22 a | 3.53 ± 0.05 b | 3.69 ± 0.11 b | 3.46 ± 0.34 b |
| (6 Z)-Nonen-1-ol | 51.14 ± 2.19 a | 52.72 ± 1.26 a | 52.68 ± 1.13 a | 52.09 ± 6.14 a | 286.45 ± 22.67 a | 249.33 ± 15.23 b | 200.71 ± 24.29 b | 180.33 ± 1.14 c | 5.91 ± 0.25 b | 6.56 ± 0.13 a | 6.02 ± 0.38 b | 6.44 ± 0.37 b |
| Subtotal | 10,163.57 ± 274.39 c | 13,691.68 ± 157.87 a | 13,524.92 ± 120.31 a | 10,883.9 ± 163.75 b | 10,873.2 ± 160.30 a | 11,808.18 ± 150.27 a | 9753.59 ± 161.16 b | 9580.37 ± 150.84 b | 12,375.14 ± 138.57 a | 8199.94 ± 115.46 c | 8750.78 ± 113.55 b | 7838.37 ± 32.42 d |
| **Straight-chain esters** | | | | | | | | | | | | |
| Acetic acid, hexyl ester | 238.53 ± 15.29 a | 253.75 ± 4.50 a | 214.13 ± 29.17 a | 171.33 ± 5.00 a | 60.23 ± 7.24 a | 74.26 ± 3.17 a | 69.54 ± 8.54 a | 68.25 ± 5.56 b | 8.64 ± 0.93 a | 5.61 ± 0.05 b | 5.92 ± 0.14 b | 4.31 ± 0.06 c |
| Acetic acid, nonyl ester | 10.79 ± 1.45 a | 8.74 ± 0.36 b | 11.25 ± 0.71 a | 15.63 ± 7.45 a | 108.17 ± 11.19 a | 123.77 ± 12.67 a | 123.89 ± 13.45 a | 109.83 ± 8.33 a | 5.82 ± 0.31 b | 6.83 ± 0.22 a | 5.22 ± 0.37 b | 5.98 ± 0.46 b |
| Acetic acid, 2-phenylethyl ester | - | - | - | - | 6608.4 ± 337.17 a | 3212.76 ± 113.67 a | 3041.27 ± 111.92 b | 2847.44 ± 12.35 c | - | - | - | - |
| Acetic acid, decyl ester | - | - | - | - | 43.2 ± 4.62 b | 34.08 ± 2.79 c | 53.05 ± 3.33 a | 41.27 ± 2.24 b | - | - | - | - |
| Acetic acid, methyl ester | 199.26 ± 10.24 b | 266.54 ± 19.26 a | 245.33 ± 11.25 a | 249.26 ± 13.32 a | 263.93 ± 12.65 a | 274.44 ± 14.22 a | 238.94 ± 13.45 a | 251.70 ± 11.25 a | - | - | - | - |
| Acetic acid, octyl ester | - | - | - | - | 448.56 ± 21.31 a | 447.3 ± 2.13 a | 400.92 ± 5.70 b | 416.23 ± 9.83 b | - | - | - | - |
| Acetic acid, pentyl ester | - | - | - | - | 266.28 ± 7.77 a | 242.5 ± 15.17 a | 256.14 ± 11.11 a | 263.52 ± 8.42 a | - | - | - | - |
| Hexanoic acid, ethyl ester | 16.89 ± 1.50 a | 15.77 ± 0.83 a | 16.46 ± 0.66 a | 18.25 ± 2.05 a | 45.29 ± 3.06 c | 197.08 ± 6.34 a | 176.13 ± 9.87 a | 96.5 ± 8.73 b | 10.52 ± 1.15 a | 6.44 ± 0.21 b | 5.96 ± 0.45 b | 6.65 ± 0.21 b |
| 3-Hexen-1-ol, acetate | - | - | - | - | 173.01 ± 12.80 a | 171.61 ± 8.08 a | 159.59 ± 8.21 a | 120.09 ± 6.35 b | - | - | - | - |
| Subtotal | 465.47 ± 28.48 b | 544.81 ± 24.95 a | 470.71 ± 41.79 b | 454.47 ± 27.82 b | 8017.07 ± 417.81 a | 4777.8 ± 178.24 b | 4519.48 ± 185.58 b | 4214.83 ± 73.06 b | 24.99 ± 2.39 a | 18.87 ± 0.48 b | 17.10 ± 0.96 b | 16.95 ± 0.73 b |
| **Straight-chain carbonyls** | | | | | | | | | | | | |
| 2-Pentanone | 115.44 ± 5.39 c | 139.35 ± 5.35 b | 93.28 ± 5.31 d | 168.53 ± 4.89 a | - | - | - | - | 14.97 ± 0.97 c | 23.23 ± 1.28 a | 23.55 ± 1.99 a | 19.32 ± 1.20 b |
| Hexanal | 30.8 ± 1.44 c | 45.25 ± 1.48 a | 37.95 ± 1.44 b | 43.45 ± 1.83 a | 51.16 ± 1.87 b | 46.03 ± 3.22 b | 41.18 ± 1.11 b | 65.39 ± 7.50 a | 17.46 ± 1.21 a | 14.86 ± 1.37 bc | 13.47 ± 0.24 c | 15.65 ± 0.45 b |
| 2-Hexenal | 452.4 ± 14.25 a | 462.23 ± 13.57 a | 485.67 ± 6.04 a | 423.54 ± 40.66 a | 174.58 ± 10.19 a | 192.2 ± 11.26 a | 190.32 ± 13.37 a | 180.02 ± 8.32 a | 494.25 ± 4.08 a | 279.24 ± 6.55 c | 415.22 ± 3.39 b | 187.84 ± 7.24 d |
| 3-Hexanone | - | - | - | - | 125.67 ± 2.32 a | 140.22 ± 3.14 a | 94.82 ± 7.44 b | 94.19 ± 2.68 b | - | - | - | - |
| 2-Octanone | 54.31 ± 1.90 b | 81.18 ± 1.32 a | 66.29 ± 1.16 a | 89.99 ± 1.61 a | 40.69 ± 3.51 a | 36.91 ± 1.55 a | 26.56 ± 1.78 b | 32.22 ± 2.22 b | 12.65 ± 4.72 a | 13.22 ± 0.52 a | 12.20 ± 1.22 a | 11.26 ± 0.80 a |
| 3-Octanone | 77.27 ± 2.44 a | 47.67 ± 0.83 b | 56.6 ± 1.94 b | 48.09 ± 0.88 b | 51.38 ± 3.35 a | 44.22 ± 2.24 b | 49.33 ± 1.11 a | 46.26 ± 1.33 b | 30.24 ± 0.49 a | 17.58 ± 1.32 c | 22.26 ± 0.33 b | 14.07 ± 1.00 d |
| Subtotal | 730.23 ± 25.42 a | 775.69 ± 22.55 a | 739.81 ± 15.89 a | 773.59 ± 49.87 a | 443.49 ± 21.24 a | 459.57 ± 21.41 a | 402.21 ± 24.81 a | 418.07 ± 22.05 a | 569.56 ± 11.47 a | 348.13 ± 11.04 c | 486.71 ± 7.17 b | 248.14 ± 10.69 d |
| **Straight-chain acids** | | | | | | | | | | | | |
| Acetic acid | 61.56 ± 3.67 b | 84.37 ± 3.27 a | 78.76 ± 1.14 b | 87.42 ± 0.62 a | 1045.01 ± 87.46 c | 1825.7 ± 26.28 a | 1450.44 ± 89.67 b | 931.88 ± 16.20 c | 10.28 ± 0.55 a | 5.30 ± 0.35 a | 9.44 ± 0.51 a | 6.62 ± 0.22 a |
| Hexanoic acid | 20.37 ± 3.09 b | 9.90 ± 0.19 c | 17.06 ± 0.29 b | 59.44 ± 1.54 a | 45.29 ± 1.06 c | 197.08 ± 1.34 a | 176.13 ± 2.87 a | 96.50 ± 1.73 b | 71.46 ± 1.50 a | 26.65 ± 1.14 b | 22.37 ± 0.32 c | 18.89 ± 0.51 b |
| Subtotal | 81.94 ± 6.76 b | 94.27 ± 3.46 b | 95.81 ± 1.43 b | 146.85 ± 2.16 a | 1090.3 ± 88.52 b | 2022.77 ± 27.62 a | 1626.58 ± 92.54 b | 1028.38 ± 17.93 b | 81.74 ± 2.05 a | 31.96 ± 1.49 b | 31.81 ± 0.83 b | 25.51 ± 0.73 b |
| Total | 11,441.21 ± 335.05 c | 15,106.44 ± 208.83 a | 14,847.71 ± 179.42 a | 12,258.81 ± 243.60 b | 20,424.06 ± 687.87 a | 19,068.33 ± 377.54 b | 16,301.85 ± 464.09 c | 15,241.65 ± 263.88 d | 13,051.44 ± 154.48 a | 8598.89 ± 128.47 c | 9286.40 ± 122.51 b | 8128.97 ± 44.57 d |

LR-BF: leaf removal at 10 days before flower, LR-AF: leaf removal at 35 days after flowers, LR-V: leaf removal at veraison. Different letters indicate significant differences ($p < 0.05$) among treatments using one-way ANOVA with the least significant different (LSD) test used to examine.

Straight-chain aliphatic esters in Cabernet Sauvignon berries were not consistent in three years. In 2017, four straight-chain aliphatic esters were detected, namely acetic acid hexyl ester, acetic acid nonyl ester, acetic acid methyl ester, and hexanoic acid ethyl ester. LF-BF significantly decreased the acetic acid nonyl ester (8.74 ± 0.36 μg/L) content compared with CN (10.79 ± 1.45 μg/L). In 2018, nine straight-chain aliphatic esters were detected. The bulk of total esters was composed of acetic acid and 2-phenylethyl ester, which decreased in all three leaf removal treatments. In addition, hexanoic acid (ethyl ester) was increased in all three leaf removal treatments, and acetic acid was increased in LR-BF (hexyl ester) and LR-AF (decyl ester). However, acetic acid (decyl ester) was decreased in LR-BF. Only three straight-chain aliphatic esters were detected in 2019. Moreover, acetic acid (hexyl ester) and hexanoic acid (ethyl ester) were decreased in all three leaf removal treatments, whereas acetic acid (nonyl ester) was increased in LR-BF.

For straight-chain aliphatic carbonyls, five carbonyls were detected in three years. The three leaf removal treatments increased 2-pentanone, hexanal, and 2-octanone concentrations but reduced the concentration of 3-octanone. However, the three leaf removal treatments had no significant effect on the total amount of straight-chain aliphatic carbonyls. In 2018, five carbonyls were detected, among which 3-hexanone and 2-octanone were decreased in LR-AF and LR-V, whereas 2-pentanone was increased in LR-V. No significant difference was found in total straight-chain aliphatic carbonyls. In 2019, five carbonyl substances were detected. Moreover, the three treatments significantly increased 2-pentanone concentration and decreased hexanal, 2-hexenal, and 3-octanone concentrations.

Two straight-chain aliphatic acids (acetic acid and hexanoic acid) were detected in three years, but the concentrations were not consistent. In 2017, LR-V increased acetic acid (87.42 ± 0.62 μg/L), hexanoic acid (59.44 ± 1.54 μg/L), and total concentration (146.85 ± 2.16 μg/L) compared with CN. In addition, high and low concentrations of acetic acid and hexanoic acid, respectively, were detected in LR-BF. In 2018, LF-BF and LF-AF increased straight-chain aliphatic acids in grape berries, whereas LF-V decreased such acid concentrations. In 2019, all three leaf removal treatments reduced the total straight-chain fatty acid concentration in berries.

### 3.4.2. Amino Acid-Derived Volatiles

Amino acid-derived volatiles could impart pleasant floral aroma, such as rose aroma, in wine [40]. The detected and identified amino acid-derived volatiles were not consistent in Cabernet Sauvignon berries during the three years, with fourteen, thirteen, and nine amino acid-derived volatiles found in 2017, 2018, and 2019, respectively (Table 5). With regard to branch-chain aliphatic acids, a high concentration of 4-methyl-2-pentyl acetate, 2-methyl-1-propanol, 4-methyl-3-penten-2-one, and 6-methyl-5-hepten-2-one was exhibited in the three leaf removal treatments to a certain degree in 2017, whereas the concentration of 6-methyl-5-hepten-2-ol decreased in LR-BF and LR-V. In 2018, LR-BF and LR-AF increased 2-ethyl-1-hexanol, 6-methyl-5-hepten-2-ol, and 6-nonen-1-ol, acetate concentrations, as well as the total concentration, whereas LR-V decreased 4-methyl-2-pentyl acetate (50.07 ± 1.14 μg/L), 3-methyl-1-butanol (180.33 ± 1.14 μg/L), and 2-ethyl-1-hexanol (2098.03 ± 16.75 μg/L) concentrations. Moreover, the concentration of 4-methyl-3-penten-2-one was decreased in all three leaf removal treatments. In 2019, the concentration of 4-methyl-3-penten-2-one was increased in the three leaf removal treatments compared with CN. However, 3-methyl-1-butanol and 2-ethyl-1-hexanol concentration were decreased in all three leaf removal treatments.

**Table 5.** Amino acid-derived volatiles in Cabernet Sauvignon berries exposed to CN, LR-BF, LR-AF and LR-V in 2017, 2018, and 2019.

| Compounds (μg/L) | 2017 | | | | 2018 | | | | 2019 | | | |
|---|---|---|---|---|---|---|---|---|---|---|---|---|
| | CN | LR-BF | LR-AF | LR-V | CN | LR-BF | LR-AF | LR-V | CN | LR-BF | LR-AF | LR-V |
| Branch-chain alphatic volatiles | | | | | | | | | | | | |
| 2-Methyl-1-propanol | 19.75 ± 1.89 c | 35.87 ± 2.27 b | 12.33 ± 0.65 d | 69.15 ± 2.17 a | 131.32 ± 3.40 a | 128.29 ± 3.17 a | 125.94 ± 6.48 a | 129.92 ± 5.13 a | - | - | - | - |
| 4-Methyl-2-pentyl acetate | 20.91 ± 2.62 c | 36.04 ± 2.83 a | 27.49 ± 1.43 b | 28.15 ± 0.54 b | 77.13 ± 1.09 a | 64.61 ± 3.89 a | 65.64 ± 1.33 a | 50.07 ± 1.14 b | - | - | - | - |
| 3-Methyl-1-butanol | 168.93 ± 13.22 a | 188.99 ± 10.52 a | 192.33 ± 9.37 a | 201.25 ± 3.56 a | 286.45 ± 22.67 d | 249.33 ± 15.21 a | 200.71 ± 14.29 a | 180.33 ± 1.14 b | 23.94 ± 1.73 a | 15.22 ± 0.68 b | 14.16 ± 1.47 b | 13.77 ± 8.77 b |
| 2-Ethyl-1-hexanol | 228.41 ± 15.41 a | 226.77 ± 1.40 a | 238.12 ± 9.69 a | 187.28 ± 2.70 b | 3410.87 ± 69.03 b | 4715.85 ± 51.29 a | 4709.98 ± 30.65 a | 2098.03 ± 16.75 c | 134.45 ± 3.47 a | 115.38 ± 1.30 a | 122.21 ± 1.46 a | 106.91 ± 3.00 b |
| 4-Methyl-3-penten-2-one | 17.7 ± 0.30 b | 36.04 ± 2.83 a | 30.49 ± 1.05 a | 32.17 ± 1.32 a | - | - | - | - | 6.45 ± 0.97 b | 15.23 ± 1.33 a | 12.36 ± 0.32 a | 13.33 ± 1.11 a |
| 4-Methyl-2-pentyl acetate | 24.24 ± 1.27 a | 21.13 ± 1.21 a | 17.49 ± 1.43 b | 19.22 ± 0.85 b | 677.13 ± 19.09 a | 464.61 ± 13.89 b | 521.09 ± 36.23 b | 501.07 ± 18.14 b | - | - | - | - |
| 6-Methyl-5-hepten-2-ol | 51.19 ± 2.39 a | 38.29 ± 1.98 b | 46.41 ± 1.61 a | 39.43 ± 0.60 b | 21.15 ± 0.41 b | 31.47 ± 1.22 a | 28.65 ± 0.93 a | 25.5 ± 0.90 b | - | - | - | - |
| 6-Methyl-5-hepten-2-one | 55.71 ± 3.88 b | 68.43 ± 1.19 a | 59.33 ± 2.35 b | 55.25 ± 1.33 b | 89.56 ± 3.45 a | 93.26 ± 3.38 a | 91.16 ± 6.26 a | 88.39 ± 2.61 a | 26.32 ± 1.23 a | 29.24 ± 1.33 a | 28.24 ± 1.22 a | 26.22 ± 1.19 a |
| 6-Nonen-1-ol, acetate | 25.44 ± 0.68 a | 27.33 ± 1.22 a | 26.33 ± 1.02 a | 24.23 ± 0.28 a | 120.39 ± 8.92 b | 139.16 ± 2.05 b | 307.14 ± 13.65 a | 113.87 ± 6.27 b | - | - | - | - |
| 1-Methyl-cyclohexene | 46.20 ± 1.41 a | 49.26 ± 2.24 a | 47.26 ± 1.26 a | 45.26 ± 1.02 a | 33.76 ± 1.05 a | 35.09 ± 0.69 a | 34.57 ± 0.98 a | 36.37 ± 0.22 a | - | - | - | - |
| Subtotal | 658.47 ± 49.54 a | 728.14 ± 27.71 a | 697.56 ± 29.85 a | 701.37 ± 14.39 a | 4847.78 ± 129.11 b | 5921.64 ± 94.78 a | 6084.88 ± 110.79 a | 3223.56 ± 59.33 b | 191.16 ± 7.41 a | 175.08 ± 4.64 a | 176.97 ± 4.47 a | 160.23 ± 14.07 a |
| Others | | | | | | | | | | | | |
| 2,4-Di-tert-butylphenol | 46.8 ± 1.70 b | 59.11 ± 2.04 a | 34.78 ± 0.91 c | 64.07 ± 1.33 a | 34.92 ± 1.65 c | 97.81 ± 1.31 a | 75.72 ± 1.57 b | 71.12 ± 1.87 b | 10.42 ± 0.25 b | 12.64 ± 0.15 a | 12.21 ± 0.51 a | 11.84 ± 0.41 a |
| Benzyl alcohol | 38.29 ± 0.16 b | 68.71 ± 1.40 a | 63.79 ± 1.48 a | 68.01 ± 2.70 a | 137.71 ± 11.19 d | 204.82 ± 10.16 b | 303.44 ± 3.68 a | 163.52 ± 8.42 c | 42.75 ± 2.96 a | 24.67 ± 0.27 c | 31.18 ± 1.84 b | 29.37 ± 1.22 b |
| Phenylethyl alcohol | 136.5 ± 4.63 c | 203.44 ± 10.56 a | 181.64 ± 8.93 a | 178.05 ± 11.22 a | 2592.75 ± 65.71 c | 5312.31 ± 54.26 b | 7584.51 ± 72.60 a | 1717.49 ± 13.38 d | - | - | - | - |
| Subtotal | 221.59 ± 6.48 c | 331.26 ± 14.00 a | 280.21 ± 11.32 b | 310.13 ± 15.25 a | 2765.38 ± 78.56 b | 5614.94 ± 65.73 a | 7963.67 ± 77.85 a | 1952.13 ± 23.67 b | 53.17 ± 3.21 a | 37.31 ± 0.42 c | 43.39 ± 2.35 b | 41.21 ± 1.63 b |
| Total | 880.06 ± 49.54 a | 1059.40 ± 41.71 a | 977.76 ± 41.17 a | 1011.50 ± 29.64 a | 7613.16 ± 207.67 b | 11,536.59 ± 160.51 a | 14,048.55 ± 188.64 a | 5175.70 ± 83.00 b | 244.33 ± 10.61 a | 212.39 ± 5.07 b | 220.37 ± 6.82 c | 201.45 ± 15.70 a |

LR-BF: leaf removal at 10 days before flower, LR -AF: leaf removal at 35 days after flowers, LR-V: leaf removal at veraison. Different letters indicate significant differences ($p < 0.05$) among treatments using one-way ANOVA with the least significant different (LSD) test used to examine.

For the other three amino acid-derived volatiles, high concentrations were found for the three treatments in 2017 and 2018, as well as the total concentration, particularly in 2018. In 2019, compared with CN, high and low concentrations of 2,4-di-tert-butylphenol and benzyl alcohol, respectively, were found in berries.

Previous studies have shown that amino acid-derived volatiles were greatly influenced by the micro-environment [41]. During berry ripening, amino acid-derived volatiles were greatly influenced by climatic conditions, precipitation, and temperature, which could promote the biosynthesis of substrate in these volatiles. In particular, leaf removal significantly elevated the temperature difference between day and night of the berry surface (Table 2), thereby improving the substances needed for the biosynthesis of amino acid-derived volatiles [33,42]. In our study, LR-BF and LR-AF greatly improved a number of amino acid-derived volatiles, which might impart pleasant floral aroma.

### 3.4.3. Isoprene-Derived Volatiles

Norisoprenoids and terpenes produced by the metabolism of isoprene mostly generated a pleasant fragrance. Although the content of isoprene-derived volatiles was low in grape fruits, their low sense threshold greatly contributed to grape aroma [43,44], which was an important component influencing the aroma of grape varieties. Two norisoprenoids and eight terpenes were detected in Cabernet Sauvignon berries (Table 6). The composition and concentration were different in three years. The three leaf removal treatments improved β-ionone, β-damascenone, and total concentrations in all three years. In particular, LR-BF showed a higher concentration.

For terpenes, the composition was not consistent in the three years. The total terpenes concentration was increased in all leaf removal treatments during the three years. However, in 2019, geraniol was not detected in all the samples, and linalool was significantly increased in LR-BF and LR-AF. Previous studies have shown that higher temperatures and stronger light were conducive to the formation of isoprene-derived volatiles [45], and increased light (Table 3) after leaf removal led to a significant increase in the content of isoprene-derived volatiles, particularly in LR-BF and LR-BF. Furthermore, shortwave light was important in the biosynthesis of norisoprenoids and terpenes (Table 3).

**Table 6.** Isoprene-derived volatiles in Cabernet Sauvignon berries exposed to CN, LR-BF, LR-AF and LR-V in 2017, 2018, and 2019.

| Compounds (μg/L) | 2017 | | | | 2018 | | | | 2019 | | | |
|---|---|---|---|---|---|---|---|---|---|---|---|---|
| | CN | LR-BF | LR-AF | LR-V | CN | LR-BF | LR-AF | LR-V | CN | LR-BF | LR-AF | LR-V |
| Norisoprenoids | | | | | | | | | | | | |
| β-ionone | 6.30 ± 0.33 b | 9.26 ± 0.30 a | 6.23 ± 0.45 b | 7.03 ± 0.57 b | 6.58 ± 0.93 b | 11.77 ± 0.38 a | 7.94 ± 0.29 b | 6.89 ± 0.33 b | 6.02 ± 0.23 b | 9.23 ± 0.13 a | 8.33 ± 0.36 a | 7.90 ± 0.65 a |
| β-damascenone | 118.99 ± 9.85 b | 159.37 ± 10.06 a | 131.07 ± 8.16 b | 127.79 ± 3.07 b | 118.27 ± 3.10 c | 159.33 ± 4.33 a | 142.91 ± 10.50 a | 128.01 ± 2.39 b | 113.74 ± 7.66 c | 182.33 ± 9.25 a | 160.21 ± 6.33 b | 140.33 ± 6.23 b |
| Subtotal | 125.29 ± 10.17 b | 168.63 ± 10.36 a | 137.29 ± 8.61 b | 134.82 ± 3.63 b | 124.85 ± 4.03 c | 171.10 ± 4.71 a | 150.85 ± 10.79 b | 134.9 ± 2.71 c | 119.76 ± 7.89 c | 191.56 ± 9.38 a | 168.55 ± 6.68 b | 148.22 ± 6.89 b |
| Terpenes | | | | | | | | | | | | |
| Geraniol | 24.35 ± 1.15 b | 41.30 ± 1.74 a | 22.63 ± 1.57 b | 16.73 ± 0.96 c | 11.97 ± 0.19 b | 47.27 ± 2.41 a | 44.26 ± 2.00 a | 42.43 ± 2.98 a | - | - | - | - |
| Linalool | 40.52 ± 1.39 b | 54.95 ± 1.85 a | 58.37 ± 1.38 a | 32.14 ± 0.63 c | 22.3 ± 0.48 c | 45.54 ± 1.16 b | 54.65 ± 0.28 a | 27.04 ± 0.46 c | 8.53 ± 0.16 b | 13.25 ± 0.66 a | 11.22 ± 0.13 a | 8.95 ± 0.23 b |
| α-Terpineol | 11.85 ± 0.25 c | 23.32 ± 0.27 a | 19.03 ± 0.34 b | 17.34 ± 0.22 b | - | - | - | - | - | - | - | - |
| Menthol | 12.68 ± 0.88 a | 10.83 ± 0.13 a | 11.22 ± 0.23 a | 11.99 ± 0.10 a | - | - | - | - | - | - | - | - |
| Citral | 38.36 ± 1.54 c | 65.26 ± 1.25 a | 46.26 ± 1.26 b | 44.26 ± 1.16 b | - | - | - | - | - | - | - | - |
| D-Limonene | 14.7 ± 0.68 c | 32.26 ± 1.26 a | 28.26 ± 0.95 b | 28.34 ± 0.62 b | - | - | - | - | - | - | - | - |
| p-Xylene | - | - | - | - | 13.61 ± 0.40 a | 14.33 ± 0.24 a | 13.56 ± 0.10 a | 13.72 ± 0.24 a | 11.17 ± 0.05 c | 13.20 ± 0.21 a | 12.21 ± 0.33 b | 11.26 ± 0.22 c |
| Furan, 2-pentyl- | 17.63 ± 0.49 b | 16.85 ± 0.69 b | 24.25 ± 0.46 a | 19.55 ± 0.47 b | 52.30 ± 1.70 | 53.25 ± 1.21 | 55.21 ± 2.00 | 55.57 ± 0.59 | 9.26 ± 0.21 b | 14.24 ± 0.22 a | 11.26 ± 0.25 b | 13.72 ± 0.56 a |
| Subtotal | 160.08 ± 6.39 b | 244.76 ± 7.17 a | 180.01 ± 6.18 b | 170.35 ± 4.16 b | 100.18 ± 2.77 c | 160.38 ± 5.02 a | 167.68 ± 4.37 a | 138.76 ± 4.27 b | 28.97 ± 0.43 b | 40.70 ± 1.10 b | 34.69 ± 0.71 b | 33.93 ± 1.01 b |
| Total | 285.37 ± 16.56 b | 413.39 ± 17.53 a | 317.30 ± 14.79 b | 305.17 ± 7.79 b | 225.03 ± 6.80 c | 331.48 ± 9.73 a | 318.52 ± 15.17 a | 273.66 ± 6.98 b | 148.73 ± 8.32 d | 232.26 ± 10.47 a | 203.23 ± 7.39 b | 182.15 ± 7.89 c |

LR-BF: leaf removal at 10 days before flower, LR -AF: leaf removal at 35 days after flowers, LR-V: leaf removal at veraison. Different letters indicate significant differences ($p < 0.05$) among treatments using one-way ANOVA with the least significant different (LSD) test used to examine.

*3.5. The Degree of Influence of Leaf Removal Treatment and Year on Volatile Aroma Compounds*

Figure 3 shows the principal component scatter plot on volatile families of compounds in grapes of leaf removal and control from 2017 to 2019. The content of all volatile aroma compounds was used as a variable, and principal component analysis was used to understand the influence of leaf removal and year on aroma substances. Principal component 1 (PC1) can explain 47.87% of the total variance, and principal component 2 (PC2) can explain 30.48% of the total variance. The cumulative contribution rate of the first two principal components has reached 78.71%. It can be seen that these two principal components are enough to explain the overall change in the variable. The first principal component can clearly separate the fruit samples of 2018 from those of the other two years. While the second principal component can separate the 2017 sample from the 2019 leaf-pickling sample, the separation of the 2018 sample is not obvious. The results showed that the aromatic substance of grape fruits was greatly influenced by year.

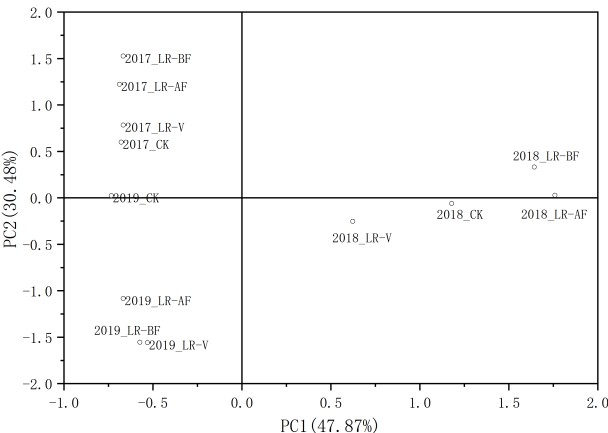

**Figure 3.** Principal component scatter plot on volatile compounds in Cabernet Sauvignon berries exposed to CN, LR-BF, LR-AF and LR-V in three years (2017, 2018, and 2019). LR-BF: leaf removal at 10 days before flower, LR-AF: leaf removal at 35 days after flowers, LR-V: leaf removal at veraison.

Figure 4 shows the corresponding loading plot on volatile families of compounds in grapes of leaf removal and control from 2017 to 2019. It can be seen from the graph that norisoprenoids, terpenes, straight-chain alphatic esters, straight-chain alphatic acids, aromatic volatiles and branch-chain alphatic volatiles in 2018 differ from that of the other two years. However, straight-chain alphatic alcohols and straight-chain alphatic carbonyls in 2017 and 2019 differ from that of 2018.

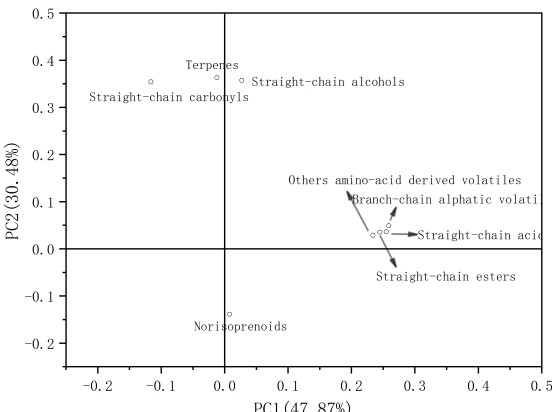

**Figure 4.** Corresponding loading plot on volatile families of compounds in Cabernet Sauvignon berries exposed to CN, LR-BF, LR-AF and LR-V in three years (2017, 2018, and 2019). LR-BF: leaf removal at 10 days before flower, LR-AF: leaf removal at 35 days after flowers, LR-V: leaf removal at veraison.

## 4. Conclusions

Leaf removal treatment at different phenological periods for three consecutive years has different effects on the physical and chemical indexes, phenol contents, and aromatic compounds of Cabernet Sauvignon grapes. Leaf removals (particularly in LR-BF) could increase TSS and phenols (anthocyanins, flavonols, and tartaric esters). Volatiles were greatly affected by time, and LR-BF could significantly increase the concentration of amino acid-derived volatiles and isoprene-derived volatiles, which could generate a pleasant flavor in *Cabernet sauvignon* grapes, followed by LR-AF and LR-V. In addition, the PCA results showed that the aromatic substance of grape fruits was greatly influenced by year. Furthermore, this study provides novel insights into leaf removal in viticulture and optimizing volatile biosynthesis at different periods in grapes. It also provides new information regarding the growth and development level of grapes.

**Supplementary Materials:** The following supporting information can be downloaded at: https://www.mdpi.com/article/10.3390/agronomy13071888/s1, Figure S1: Photos of leaf removals.

**Author Contributions:** Conceptualization, Z.L.; methodology, D.Y.; software, X.G.; validation, X.G. and Y.S.; funding acquisition, X.G., Y.S. and J.W. All authors have read and agreed to the published version of the manuscript.

**Funding:** This work was financially supported by Natural Science Foundation of Shandong Province (ZR2022MC168), Special Fund for International cooperation of Shandong Academy of Agricultural Sciences (CXGC2022F27), Scientific Research Guidance Fund of Shandong Academy of Grape (Grant No: SDAG2021B01), Major Project of Science and Technology of Shandong Province (Grant No: 2022CXGC10605), the National Natural Science Foundation of China (Grant No: 31701887), Key Areas Science & Technology Project of the 12th Division, Xinjiang Production and Construction Corps (Grant No: SRS2022034), and Agricultural Science and Technology Innovation Project of Shandong Academy of Agricultural Sciences (Grant No: CXGC2023F15).

**Data Availability Statement:** Not Applicable.

**Acknowledgments:** Throughout the study and during the manuscript preparation, the project team received tremendous administrative and logistical support from the management of the Shandong Academy of Agricultural Sciences and Cofco Great Wall Wine (Yantai) Co., Ltd., Yantai, China for which the authors are very grateful.

**Conflicts of Interest:** The authors declare no conflict of interest. The funders had no role in the design of the study; in the collection, analyses, or interpretation of data; in the writing of the manuscript; or in the decision to publish the result.

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
