# Peer review of "Changes in Volatile Composition of Cabernet Sauvignon (Vitis vinifera L.) Grapes under Leaf Removal Treatment"

_agronomy, doi:10.3390/agronomy13071888_

Round 1

Reviewer 1 Report

Dear Authors,

The paper s topic is interesting and actual and the authors improved the manuscript.

The authors present information about the general climatic condition of the Yantai region (geographical coordinate, mean temperature, max/min temperature, useful/active temperatures, sunlight hours, and so on).

Line 66-67 – The following phrase ~The vines were single cordon trained and 8-12 fruiting branches were retained~ it is hard to understand...Please indicate the number of buds left on the vine after pruning. Also, use an appropriate word for `fruiting branches` as `canes`. Thank you!

Figures 1 and 2 – please indicate the statistical test used

Subchapter 3.3. – it is possible to state a clear cause of the lower amount of anthocyanins, tartaric esters, and total phenols in 2018 and 2019 than in 2017?

In the Results section, the most significant results must be exposed as data. It is not appropriate to only state that some compounds are increasing/decreasing and referring the number in the Table where presented. Also, the results must be compared with other studies on the identical/similar topic (follow the  Toni Kujundži´ et al., 2022)

Thank you!

Author Response

Response to Reviewer 1 Comments

Point 1: The paper s topic is interesting and actual and the authors improved the manuscript.

The authors present information about the general climatic condition of the Yantai region (geographical coordinate, mean temperature, max/min temperature, useful/active temperatures, sunlight hours, and so on).

Response 1: We sincerely thank the reviewer for his interest and positive comments on our research.

Point 2: Line 66-67 – The following phrase ~The vines were single cordon trained and 8-12 fruiting branches were retained~ it is hard to understand...Please indicate the number of buds left on the vine after pruning. Also, use an appropriate word for `fruiting branches` as `canes`. Thank you!

Response 2: Thanks for your useful comment. We have revised the phrase in the revision as “The vines were single cordon trained and 8-12 two-bud-spurs per vine were retained. The vineyard soils were clean tillaged, and organic fertilizer was applied every winter and spring.” in Line 66-68

Point 3: Figures 1 and 2 – please indicate the statistical test used

Response 3: Thanks for your useful comment. We have indicated the statistical test used in the revision as “Different letters indicate significant differences (P < 0.05) among treatments using one-way ANOVA with the least significant different (LSD) test used to examine.” in Line 200-202 and Line 219-220.

Point 4: Subchapter 3.3. – it is possible to state a clear cause of the lower amount of anthocyanins, tartaric esters, and total phenols in 2018 and 2019 than in 2017?

Response 4: Thanks for your careful work. According to literature reports, sunlight had a positive effect on the synthesis of phenols in grapes, In our study, the total sunshine duration was 873.8, 826.4, and 848.6 h in this three years from July to October (veraison to harvest) according to the Subchapter 3.1, and which means that the grapes received the most total sunshine duration during the ripening process in 2017. That explains the lower amount of anthocyanins, tartaric esters, and total phenols in 2018 and 2019 than in 2017. We have revised it in the revision in Line 214-218.

Point 5: In the Results section, the most significant results must be exposed as data. It is not appropriate to only state that some compounds are increasing/decreasing and referring the number in the Table where presented. Also, the results must be compared with other studies on the identical/similar topic (follow the Toni Kujundži´ et al., 2022)

Response 5: Thanks for your useful suggestion. We have revised “Subchapter 3. Results and Discussion” of the manuscripts. For example, Lines 157-164, Lines 190-194, Lines 212-221, and so on. All the changes was marked in red. In addition, the Conclusion have also been revised to make it more accurate.

Reviewer 2 Report

I do not have any comments.

The article can be published in its current form.

Author Response

Point 1:I do not have any comments.

The article can be published in its current form.

Response 1: Thank you for your recognition of our work. We have carefully examined the manuscript again and revised some contents to make the paper more rigorous and scientific.

The revised manuscript was attached, please see the attachment.

Reviewer 3 Report

Comments to the Author:

This manuscript studies the volatile composition of Cabernet Sauvignon grapes under leaf removals. The paper is original and interesting but some changes must be done before its possible publication. In general authors show too many results and a poor discussion about them. In my opinion, the environment tables must be changed to the Supplementary material. On the other hand I consider that phenolic composition is not fitted with the title of the paper. The authors has too many information of the volatile composition, so it is not necessary to include phenolic composition in this manuscript.

Some comments:

-Lines 78-80: In these lines the authors put that after freezing the grapes they analysed the soluble sugars, the pH and the total acidity, and then in general parameters as well. Please clarify this in the text.

-Line 104: What standards were used to quantify the rest of the phenolic families?

-Line 117: What´s the meaning of “no shunt model was introduced”?

-Line 119: What´s test was used to differentiate means?

-Line 224: Change “32” for “Thirty two”

-In general, in the legent tables, change "according to ANOVA" to "according to....." the corresponding test.

- In this part of the  manuscriptThe degree of influence of leaf-removal treatment and year on volatile aroma compounds”, authors explain a principal component analysis and in the part of “Statistical analysis”, they did not name nothing about it. On the other hand, Figure 4 did not correspond to the explained text.

In short, in my opinion major changes must be done in the manuscript before its publication in this journal

Author Response

Point 1:This manuscript studies the volatile composition of Cabernet Sauvignon grapes under leaf removals. The paper is original and interesting but some changes must be done before its possible publication. In general authors show too many results and a poor discussion about them. In my opinion, the environment tables must be changed to the Supplementary material. On the other hand I consider that phenolic composition is not fitted with the title of the paper. The authors has too many information of the volatile composition, so it is not necessary to include phenolic composition in this manuscript.

Response 1:Thanks a lot for carefully reviewing our work. The comments were very helpful for revising and improving our paper and a meaningful guide to our other research. We have studied the comments carefully and made corrections in the manuscript. Although our manuscript focused on the study of leaf removal on volatile composition, we also focus on other quality indicators of the grapes. because leaf removal not only affects the volatile composition of the grapes, but also other secondary metabolites such as phenolic composition after its leaf being removed, which also have an important impact on the quality of the wine. In addition, the study of phenolics is the next step to focus on.

Point 2: -Lines 78-80: In these lines the authors put that after freezing the grapes they analysed the soluble sugars, the pH and the total acidity, and then in general parameters as well. Please clarify this in the text.

Response 2: Thanks a lot for your careful opinion. We have put the description related to "sample preparation" into Section 2.2 "General Grape Parameters" in Lines 79-80 to make the description clearer.

Point 3: -Line 104: What standards were used to quantify the rest of the phenolic families?

Response 3: Thanks for your question. We have revised the comments in the revision to " The calibration curve of total phenols, tartrate, flavonols and anthocyanins were obtained using 0 ~ 100 μg/mL gallic acid, caffeic acid, quercetin and malvacin-3-glycoside, respectively." in Lines 100-102.

Point 4: -Line 117: What´s the meaning of “no shunt model was introduced”?

Response 4: Thanks a lot for your careful work. We have removed this irrelevant phrase in the revision.

Point 5: -Line 119: What´s test was used to differentiate means?

Response 5: Thanks for your useful work. A one-way ANOVA with the least significant different (LSD) test used to examine was performed using SPSS 20.0 program (SPSS, USA) to detect significant differences (P < 0.05) among leaf removal treatments at each year for all parameters considered. We have revised this comments in Lines 145-148.

Point 6: -Line 224: Change “32” for “Thirty two”

Response 6: Thanks for your useful suggestion. We have changed the numb "32" to " Thirty two ", and the other similar numbers have been revised as well.

Point 7: -In general, in the legent tables, change "according to ANOVA" to "according to....." the corresponding test.

Response 7: Thanks for your careful work. We have revised it to " Different letters indicate significant differences (P < 0.05) among treatments using one-way ANOVA with the least significant different (LSD) test used to examine. "

Point 8: - In this part of the  manuscript “The degree of influence of leaf-removal treatment and year on volatile aroma compounds”, authors explain a principal component analysis and in the part of “Statistical analysis”, they did not name nothing about it. On the other hand, Figure 4 did not correspond to the explained text.

Response 8: Thanks for your careful suggestion. We have revised the description " Factor analysis was performed (SPSS) to determine the relationship among different variables and select the principal component analysis (PCA) analysis to indicate the relationship between volatile compounds and leaf removing in different years." in lines 148-151 at " Statistical analysis ". In addition, we have check Figure 4 carefully, and revised the correct picture and explained text.

Point 9: In short, in my opinion major changes must be done in the manuscript before its publication in this journal.

Response 9: Thank you again for your careful work and suggestion. We have revised “Subchapter 3. Results and Discussion” of the manuscripts. For example, Lines 157-164, Lines 190-194, Lines 212-221, and so on. All the changes was marked in red. In addition, the Conclusion have also been revised to make it more accurate.

The revised manuscript was attached, please see the attachment.
